# Knowledge and Practice of Health Professionals in the Management of Dysphagia

**DOI:** 10.3390/ijerph18042139

**Published:** 2021-02-22

**Authors:** Eduardo Sánchez-Sánchez, Ylenia Avellaneda-López, Esperanza García-Marín, Guillermo Ramírez-Vargas, Jara Díaz-Jimenez, Francisco Javier Ordonez

**Affiliations:** 1Internal Medicine Department, Punta de Europa Hospital, Algeciras, 11207 Cádiz, Spain; ain3ly@gmail.com (Y.A.-L.); espe_garciamarin@hotmail.com (E.G.-M.); guiram1992@gmail.com (G.R.-V.); 2Faculty of Education Sciences, University of Cádiz, 11519 Puerto Real, Spain; luna_nueva17@hotmail.com; 3Human Anatomy, School of Medicine, University of Cádiz, Plaza Fragela s/n, 11003 Cádiz, Spain; franciscojavier.ordonez@uca.es

**Keywords:** bronchoaspiration, dysphagia, MECV-V, pneumonia, health professionals

## Abstract

The aim of this study was to determine healthcare providers’ knowledge and practices about dysphagia. A descriptive cross-sectional study was carried out based on a self-administered and anonymous questionnaire addressed to healthcare providers in Spain. A total of 396 healthcare providers participated in the study. Of these, 62.3% knew the definition of dysphagia as a swallowing disorder. In addition, up to 39.2% of the participants reported that they did not know whether the EatingAssessmentTool (EAT-10) dysphagia screening test was usedin their own clinical settings. Similarly, up to 49.1% of them did not know the ClinicalExaminationVolume-Viscosity (MECV-V) method. Nearly all participants (98.8%) reported that thickeners must be used forall liquids administered to patients. A higher percentage of respondents based the choice of texture on patient’s tolerance (78.2%) rather than on the MECV-V result (17.3%). In addition,76.4% of the professionals had witnessed a bronchoaspiration; after it, 44.4% (*n* = 175) of them reported the appearance of pneumonia, and 14.5% (*n* = 57) the death of the patient (*p* = 0.005). The participants revealeda moderate/low knowledge ofthe definition, diagnosis, and clinical management of liquid dysphagia, which indicates some room for improvements.

## 1. Introduction

Nutrition is a basic function of the living being, carried out involuntarily for the acquisition of the necessary nutrients for the correct functioning of the organism. Nutrition starts with the processing of food in the mouth, which involves chewing, formation of abolus, and swallowing, and continues with the bolus being transported to the digestive tract where digestion and subsequent absorption of the different nutrients will take place. It is for this reason that swallowing is an important step in obtaining the body’s requirements and needs in a safe and effective way [1]. The digestive tract runs parallel to the respiratory one, and the two share anatomical structures; therefore, swallowing must be carried out in a coordinated manner, so that food cannot enter the airway [2].

Dysphagia or impaired swallowing refers to difficulties in any of the three main phases of the swallowing process, i.e., the oral, pharyngeal, and oesophageal phases [3]. Dysphagia can appear with solid or liquid intake.

It is widely accepted that it is a prevalent disorder among older adults. In fact, signs of dysphagia were common among patients aged 65 years or older in acute care settings [4,5]. The current situation could be even worse in the institutionalized elderly population [6]. Swallowing inefficiency could also be prevalent in younger individuals with amyotrophic lateral sclerosis [7].

The consequences of fluid dysphagia include tracheobronchial aspiration [8]. Although some clinical signs, like coughing, help in the diagnosis of aspiration, silent aspiration may occur [9], which, being asymptomatic, may remain undetected by observation [10]. This may lead to aspiration pneumonia [8]. Pneumonia is one of the main causes of mortality after a stroke [11,12]. Dysphagia patients have a higher rate of pneumonia compared to those without dysphagia (29.7% vs. 3.7%). Dysphagia can reduce patients’ quality of life and increase hospital stays, socio-sanitary charges, and the risk of mortality [13,14,15]. Limited food and fluid intake, as well as abuse of fast to avoid aspiration in patients with liquid dysphagia, are risk factors for the onset of malnutrition and dehydration [16].

The care of patients with liquid dysphagia aims to reduce the complications derived from this condition. It is recommended to adopt some general measures. These measures include the adaptation of food texture and viscosity according to the degree of dysphagia present, avoiding foods with two textures, especially for patients with liquid dysphagia, avoiding the use of syringes and of straws for the oral administration of liquids, as they do not stimulate swallowing and promote aspiration, placing and maintaining the patient in a suitable position before and during ingestion, adapting the kitchenware to the patient’s needs [17].

Liquid dysphagia is one of the most underdiagnosed disorders by health professionals [1]. The lack of diagnosis is associated with negative outcomes, which leads to the increase of hospital stays, socio-health costs, and one-year mortality [15,18,19]. For dysphagia patients, the hospital costs directly associated with dysphagia amount to 3677 euros, and those associated with home care to 6192 euros [20].

Clinical bedside assessment may be helpful for the diagnosis of dysphagia and its severity; however silent aspirations cannot be detected. For the diagnosis of silent dysphagia, tests must be used, being the Video fluoroscopic swallowing studies the gold standard for its diagnosis [21]. There are different clinical methods for the screening and diagnosis of liquid dysphagia, among which the screening methods based on the Eating Assessment Tool (EAT-10) and on the Clinical Examination of Volume Viscosity (MECV-V) [22]. The EAT-10 is a simple and fast method that evaluates the presence of specific symptoms of dysphagia and has been shown to have high internal consistency and reproducibility [23].

The MECV-V is a valid and reliable screening test [24]. It has high sensitivity and specificity [25] for the detection of dysphagia [26]. It is based on a detailed protocol designed to maintain the safety and efficacy of swallowing and detect silent aspirations [27]. It allows establishing the diagnosis of dysphagia and guiding dietary adaptations so that swallowing becomes safe and effective. This method evaluates the presence of three different types of food texture on the basis of their viscosity—i.e., liquid, nectar, and pudding—in three volumes (5 mL, 10 mL, and 20 mL) [28]. Changes in any of the signs associated with safety (voice tone, cough, or oxygen desaturation) or efficacy (lip seal, oral residue, fractional swallowing, and pharyngeal residue) label the test positive [22].

Recent studies have pointed out that a multidisciplinary team approach was important to properly identify and manage dysphagia [29,30]. However, the full team (doctors, physical therapists, speech therapists, rehabilitators, nurses, auxiliary nursing care technicians (ANCTs)) may not be available in all clinical settings, and thus isolated healthcare providers should deal with dysphagia. As a consequence, the knowledge of these professionals about dysphagia management are essential for the appropriate continuing professional development in this area [29].

This study was aimed to determine the knowledge and practices of Spanish healthcare providers (nurses, doctors, ANCTs) who care for patients with liquid dysphagia or at risk of suffering from it. This information may play a key role when preparing future clinical training programs and detailed algorithms for the clinical management of dysphagia.

## 2. Materials and Methods

A cross-sectional descriptive study based on an anonymous and self-administered questionnaire was conducted for nurses working in any field of health. The current online medical questionnaire was administered on a free platform (Google) [31]. A non-probabilistic or convenience sample was used, since the questionnaire was open to all nurses, doctors, and ANCTs who carried out their healthcare activity in Spain.

In order to estimate the sample size, we considered the last report of the National Health System established in 2017, which stated that there were 245,533 nurses and 178,600 doctors in Spain [32]. For ANCTs, figures were not included, but a total of 120, 110 has been estimated [33]. Accordingly, a total of 544,243 health professionals worked in Spain in 2017. Accordingly, the current sample size (*n*= 384) was calculated assuming 95% confidence level or 5% level of significance and a 10% margin of error.

No standardized questionnaire on healthcare providers´ knowledge and practices about dysphagia was found in the literature. Therefore, we designed a semi-structured questionnaire, based on the doubts and questions of healthcare professionals who had access to training courses delivered nationally by the authors in the last 2 years. A pilot study was carried out among colleagues from our unit and others who had received training to check the effectiveness of the questionnaire, in order to know if it provided us with the necessary information and if we should modify any questions. After this, questions were modified when they were not sufficiently understandable, which could lead to an interpretation error.

The first part of the questionnaire included sociodemographic variables such as age, sex, province of residence, and unit where the professionals carry out their activity. The second part included questions about diagnosis, management, recommendations, and complications regarding dysphagia. Lastly, in order to guarantee the dissemination of the questionnaire among Spanish healthcare providers, social platforms such as Twitter, Facebook, WhatsApp, and Instagram were used. The questionnaire was administered between March and June 2019.

All data obtained from the questionnaire were represented descriptively. The qualitative variables were their frequency and percentage, and the quantitative variables were expressed by the mean and the standard deviation or dispersion. Subsequently and by the use ofthe Chi-square test, significant differences between different groups were assessed, assuming a confidence level of 95%. Significance was set at an alpha level of 0.05. Statistical treatment was conducted using the free R-Commander package.

## 3. Results

A total of 400 responses to the questionnaire were obtained, but 6 were deleted because of missing data for some responses. Of the 396 subjects studied, 82.5% were women (*n* = 325), and 17.5% were men (*n* = 69). The mean age of the respondents was 39.4 ± 9.3 years (39.1 ± 9.5 years for women; 40.4 ± 8.1 years for men).

The professional category with the highest representation in the sample was nursing staff, corresponding to82.2% (*n* = 324), followed by ANCT (13.9%, *n* = 55) and doctors (3.8%, *n* = 15) (*p*<0.001).

As the data collection was carried out through social network platforms, participants responded from the whole country (Spain). With regard to Autonomous Communities, Andalusia was the most represented, with 44.6% (*n* = 176) of participants, followed by the Balearic Islands (12.4%, *n* = 49), Madrid (10.1%, *n* = 40), and Catalonia (6.3%, *n* = 25). The least represented were Cantabria and Murcia, with 0.76% (*n* = 3) and 0.3% (*n* = 1) of participants, respectively.

Regarding the assistance units or departments, the most represented in the current study were Internal Medicine (*n* = 97; 24.6%), Geriatrics (*n* = 43; 10.9%), and Primary Care (39; 9, 9%), with the professionals of the Social Health Centers being the least prevalent (*n* = 6; 1.5%). The other category included units such as urology, cardiology, pneumology, etc. Of the subjects surveyed, 45.7% (*n* = 180) answered that there were Nutrition Units in their centers, whereas 18.0% did not know if these units were present in their centers.

Up to 63.2% (*n* = 249) of the respondents reported that they knew the definition of dysphagia, there being no statistically significant differences between the different professional categories (*p* = 0.350). A contingency table was defined for two variables, one of them being the unit of work, and the other the knowledge of the definition of dysphagia. The results showed that a higher percentage of emergency professionals (*n* = 21; 87.5%), than of those who worked in socio-health centers (*n* = 1; 16.7%) defined dysphagia, with differences statistically significant between the different units or services (*p* < 0.001). There were no statistically significant differences in the definition of dysphagia between the professionals when considering the presence or absence of Nutrition Units in their centers (*p* = 0.107) (Table 1).

The data obtained on health professionals’ knowledge and management of liquid dysphagia is detailed by groups in Table 2. The approach to a patient with dysphagia through the performance of a test was the most prevalent response (57.9%, *n* = 228), with respect to the prohibition of fluid intake (*n* = 12; 3.0%) and the application of absolute diet guidelines (*n* = 5; 1.3). Conversely, 37.8% (*n* = 149) of respondents reported not knowing which approach should be taken with these patients. No significant differences were found between groups (*p* = 0.130).

Another question asked was whether screening tests and the diagnosis of liquid dysphagia were carried out in their clinical settings. Up to 20.8% (*n* = 82) of the subjects reported that the EAT-10 screening test was performed in their centers, whereas 39.2% (*n* = 154) of the participants did not know the test, with no differences by category of professionals (*p* = 0.136). When faced with the same question in relation to the MECV-V test, 24.1% (*n* = 95) of the respondents answered that such a test was performed in their clinical setting, and 41.9% (*n* = 165) reported not knowing it. There were statistically significant differences between the different categories of professionals (*p* = 0.008), as 52.7% (*n* = 29) of the ANCTs did not know the test compared to 26.7% (*n* = 4) of the doctors and 40.7% (*n* = 132) of the nurses.

Up to 41.4% (*n* = 163) of the respondents answered that these tests were performed on the population at risk, and only 3.3% (*n* = 13) on patients after suffering a bronchoaspiration, with no differences in the responses among the different professionals (*p* = 0.382). When asked if there were subsequent controls for dysphagia, 166 (42.1%) professionals answered positively, and 163 (41.4%) did not know. They were asked who was in charge of performing this test, to which questions, 37.3% (*n* = 147) of the participants answered that they did not know. Among those who knew the person in charge, the most prevalent response (35.0%, *n* = 138) was the nurse, and the respondents who provided this answer were more frequently nurses (*n* = 117; 36.1%) and ANCTs (*n* = 19; 34.5%), than doctors (*n* = 2; 13.3%), although there was no significant difference between professional categories (*p* = 0.114)

The results obtained for the management of thickeners showed that 98.8% (*n* = 389) of the participants reported that thickeners should be used in all liquids, and 1.2% (*n* = 5) of them answered that they should be used in some of the main meals (breakfast, lunch, dinner), although there were no significant differences between professionals (*p* = 0.317). A high percentage of professionals based their choice of texture on patient tolerance (*n* = 308; 78.2%), with palatability being the factor on which their decision was least based (*n* = 18; 4.5%). On the other hand, 17.3% (*n* = 68) of the respondents based their choice on the results of a previous MECV-V test (*p* = 0.497).

Regarding the recommendations that had to be followed for the management of a patient with dysphagia, 92.6% (*n* = 365) of the participants reported that they always adjusted a patient’s position before ingestion. There were statistically significant differences between the different categories (*p* < 0.001), being the percentage of ANCTs that accommodate patients higher than those of nurses and doctors (98.2% vs. 92.9% vs. 66.7%, respectively). In addition, differences were observed in the use and/or recommendation of straw and syringes for the administration of liquids among professionals (ANCT: 76.4%, nurse: 58.6%, and doctor: 60.0%, *p* = 0.044). In addition, 81.9% (*n* = 323) of the respondents always provided information to relatives about preventive measures and the management of aspiration (*p* = 0.093).

Lastly, 23.6% of the professionals surveyed (*n* = 93) had not witnessed a bronchoaspiration. Among those respondents who had witnessed it, 18.5% (*n* = 73) reported that, after that episode, the patient had no incidents, 44.4% (*n* = 175) replied that the patient had presented pneumonia, and 14.5% (*n* = 57) answered that aspiration caused the death of the patient. Significant differences were found among different professional categories (*p* = 0.005).

## 4. Discussion

The management of liquid dysphagia plays an important role in the health care provided by different health professionals and aims to reduce or avoid consequences derived from this dysfunction.

In the study published by Far pour et al. in 2019 [34], it was observed that 96.82% of the health personnel interviewed knew the definition of dysphagia as a swallowing disorder. This value is higher than that reported in our study (63.2%). The aforementioned study was carried out in three university hospitals in three major cities in Iran, so the sample consisted of professionals from three large hospitals, whereas our sample was more dispersed because the professionals surveyed were carrying out their work throughout the national territory, that presents differences in the health system and in the protocols for the management of patients with dysphagia.

The lack of knowledge of the term dysphagia can lead to unjustified practices in these patients, such as the use of fasts or the prohibition of fluid intake, or to indecision about what approach to choose, thus increasing the negative consequences of dysphagia [35].

Although the data on the prevalence of dysphagia reported in the introduction to this article indicated a very high value in the institutionalized elderly population as well as in patients with different neurological diseases, the percentage of health personnel interviewed, working in Neurology departments, who knew the definition of dysphagia was close to 50%, and that of professionals working in health centers was 16.7%. However, the percentage of professionals working in units such as the Emergency department was high, possibly due to the fact that the lack of an adequate diagnosis makes patients go to the Emergency department more often to be treated for the negative consequences of dysphagia.

The early diagnosis of dysphagia helps healthcare professionals to direct care to minimize its risks and consequences. The results obtained showed that only 20.8% of the subjects in our study stated that the EAT-10 dysphagia screening test was performed in their center, this value being very close (24.1%) to the percentage of participants who stated that the MECV-V scanning method was performed. These figures are lower than those reported by Farpour et al., who concluded that between 49.9% and 52.2% of the participants in their study had used a method to evaluate or treat dysphagia [35]. These tests should be performed on the entire population at risk, so a higher percentage than that reported in our study (41.4%) should be achieved.

In the review carried out by Hines et al., it was shown that the detection of dysphagia by nurses improves the management of patients with dysphagia [36]. In our study, a high percentage of nurses declared to conduct the tests, which cabimprove the approach to these patients in our territory.

Thickeners help achieve the texture or viscosity that allows a safe swallowing [37]. The respondents used thickeners in all liquids, and the texture was chosen in relation to the tolerance of the patient, without carrying out an adequate diagnostic test (MECV-V).

Among the recommendations for the management of patients with dysphagia are postural considerations [38]. A high percentage of professionals’ report that they adjusted the position of patients with dysphagia before eating, and this percentage was higher for ANCTs and nurses. This may be due to the fact that these professionals provide bedside care, which includes the effective and efficient management of the patient’s oral route while eating. Sometimes, these professionals are in charge of feeding patients who cannot eat autonomously.

In the protocol published by García et al. in 2018, entitled “Protocol for diagnosis and treatment of oropharyngeal dysphagia in the elderly”, it can be seen how the use of straw and syringes should be avoided in elderly patients with dysphagia to prevent aspiration during fluid intake. Our results suggested that a higher percentage of ANCTs provided straws and syringes compared to the nursing staff [39].

The relatives of these patients should be provided with information on preventive measures and management of bronchoaspiration, in order to empower them in the care of their relatives. In our survey, 81.9% of the respondents reported that they informed family members, but the results described above suggest that this information might be erroneous and, rather than bringing benefits to the patient and the system, it would have consequences negative.

Tracheobronchial aspirations can cause frequent respiratory infections. Up to 50% of patients with dysphagia can develop aspiration pneumonia, with an associated mortality of up to 50% [40,41]. Our results show that 76.4% of the subjects reported having witnessed an episode of aspiration, but 18.5% of them reported that this episode did not cause any incidence. It should not be forgotten that silent aspiration is one of the main complications that patients present. In this study, 44.4% of the respondents reported that aspiration resulted in pneumonia, and 14.5% of them that this pneumonia led to the death of the patient. If we take into account the relative frequencies, that is, with respect tothe total number of aspirations observed, 58.1% of them resulted in pneumonia, and 32.6% of these pneumonia occurrences resulted in the death of the patient.

The lack of training of health professionals who provide their services to patients at risk of or with oropharyngeal dysphagia can lead to a delay in patients’ diagnosis and increase the complications derived from this condition, which is an important barrier to the management of these patients [42]. Several authors have studied the knowledge possessed by health professionals, mainly nurses, finding that this knowledge was moderate [43] and that specific training and experience in caring for patients with dysphagia provided new and better knowledge [29].

Although the number of responses exceeded the calculated sample size, it is difficult to infer these results, because this study used a convenience sampling rather thana probability sampling. Furthermore, there was a disparity between the number of participants per category and the different geographical areas they came from.

Further research in this field should be carried out, a validated questionnaire developed, and this questionnaire should be presented before and after training, to verify the effectiveness of the training.

## 5. Conclusions

The results of the present study showed that the participants had a moderate/low knowledge of the definition, diagnosis, and clinical management of liquid dysphagia, so necessary tools must be provided for their training in this field. This training must be multidisciplinary and should be directed to all professionals who provide healthcare to patients with liquid dysphagia.

This study shows the need for the implementation of guidelines and/or protocols for the management of patients with dysphagia, with the aim of promoting the training of different health professionals. In addition, it highlights the training needs of these professionals with respect to dysphagia to improve their approach to patients with this condition and allow them to identify the signs of dysphagia, so to refer patients to a qualified specialist.

## Figures and Tables

**Table 1 ijerph-18-02139-t001:** Work units and presence of Nutrition Units.

	*n*	%	Definition of Dysphagia		*p*-Value
			Impaired Swallowing (*n*,%)	Do Not Know	
				*n* (%)	
Reference Units:					<0.001
Surgical area	20	5.5	12 (60.0)	8 (40.0)	
Primary Health Care	39	9.9	30 (6.9)	9 (23.1)	
Socio-Health centers	6	1.5	1 (16.7)	5 (83.3)	
Critical Care Medicine	26	6.6	16 (61.5)	10 (38.5)	
Palliative Care	10	2.6	8 (0.0)	2 (20.0)	
Geriatrics	43	10.9	34 (79.1)	9 (20.9)	
Internal Medicine	97	24.6	49 (50.5)	48 (49.5)	
Neurology	15	3.8	7 (46.7)	8 (53.3)	
Oncology	31	7.9	14 (45.2)	17 (54.8)	
Otorhinolaryngology	7	1.8	5 (71.4)	2 (28.6)	
Emergency	24	6.1	21 (87.5)	3 (12.5)	
Others	76	19.3	52 (68.4)	24 (3.6)	
**Nutrition Unit in the center:**					
Yes	180	45.7	123 (54.9)	57 (45.1)	
Not	143	36.3	87 (60.8)	56 (39.2)	0.107
Do not know	71	18	39 (68.3)	52 (31.7)	

**Table 2 ijerph-18-02139-t002:** Health professionals’knowledge and management of patients with liquid dysphagia.

Questionnaire/Answer to the Questionnaire	Nurse (N = 324)	Doctor (N = 15)	ANCT (N = 55)	Total (N = 394)	*p*-Value
	n (%)	n (%)	n (%)	n (%)	
**What is the definition of dysphagia?**					
Impaired swallowing	210 (64.8)	8 (53.3)	31 (56.4)	249 (63.2)	0.35
Do not know	114 (35.2)	7 (46.7)	24 (43.6)	165 (36.8)	
**What approach would you take witha patient with liquid dysphagia?**					
Dysphagia test and adaptation of the diet	194 (59.9)	8 (53.3)	26 (47.3)	228 (57.9)	
No fluid intake	8 (2.5)	0 (0.0)	4 (7.3)	12 (3.0)	
Absolute diet	3 (0.9)	1 (6.7)	1 (1.8)	5 (1.3)	0.13
Do not know	119 (36.7)	6 (40.0)	24 (43.6)	149 (37.8)	
**Is the EAT-10 test performed at your center?**					
Yes	64 (19.8)	3 (20.0)	15 (27.3)	82 (20.8)	
Not	73 (22.5)	2 (13.3)	4 (7.3)	79 (20.0)	0.136
I do not know the test	127 (39.2)	5 (33.3)	22 (40.0)	154 (39.2)	
Do not know	60 (28.5)	5 (33.3)	14 (25.5)	79 (20.0)	
**Is the MEDCV-V test performed at your center?**					
Yes	83 (25.6)	3 (20.0)	9 (16.4)	95 (24.1)	
Not	57 (17.6)	2 (13.3)	2 (3.6)	61 (15.5)	0.008
I do not know the test	132 (40.7)	4 (26.7)	29 (52.7)	165 (41.9)	
Do not know	52 (16.0)	6 (40.0)	15 (27.3)	73 (18.5)	
**When are these tests performed?**					
Upon admission	46 (14.2)	2 (14.2)	11 (20.0)	59 (15.0)	
On request	43 (13.3)	4 (26.7)	4 (7.3)	51 (12.9)	
With people at risk of dysphagia	136 (42.0)	6 (40.0)	21 (38.2)	163 (41.4)	0.382
After bronchoaspiration	9 (2.8)	0 (0.0)	4 (7.3)	13 (3.3)	
Do not know	90 (27.8)	3 (20.0)	15 (27.3)	108 (27.4)	
**Are subsequent checks carried out?**					
Yes	136 (42.0)	6 (40.0)	24 (43.6)	166 (42.1)	
Not	53 (16.4)	4 (26.7)	8 (14.5)	65 (16.5)	0.85
Do not know	135 (41.7)	5 (33.3)	23 (41.8)	163 (41.4)	
**Who carries out these tests?**					
Doctor	21 (6.5)	4 (26.7)	8 (14.5)	3 (0,8)	
Nurse	117 (36.1)	2 (13.3)	19 (34.5)	138 (35.0)	
Speech therapist	8 (2.5)	0 (0.0)	0 (0.0)	8 (2.0)	
Nutrition unit	46 (14.2)	4 (26.7)	11 (20.0)	61 (15.5)	0.114
Other	4 (1.2)	0 (0.0)	1 (1.8)	5 (1.3)	
Do not know	126 (38.9)	5 (33.3)	16 (29.1)	147 (37.3)	
Not applicable	2 (0.6)	0 (0.00)	0 (0.0)	2 (0.5)	
**If the patient requires thickeners, when is the thickener used?**					
Breakfast	0 (0.0)	0 (0.0)	1 (1.8)	1 (0.2)	
Lunch	3 (0.9)	0 (0.0)	0 (0.0)	3 (0.8)	0.317
Dinner	1 (0.3)	0 (0.0)	0 (0.0)	1 (0.2)	
In all fluids the patients ingest	320 (98.8)	15 (100.0)	54 (98.2)	389 (98.8)	
**Based on whatdo you choose the texture to be prescribed topatients with liquid dysphagia?**					
MECV-V test results	59 (18.2)	3 (20.0)	6 (10.9)	68 (17.3)	
Patient tolerance	249 (76.9)	11 (73.3)	48 (87.3)	308 (78.2)	0.497
Palatability	16 (4.9)	1 (6.7)	1 (1.8)	18 (4.5)	
**Do you adjust the position ofapatient with liquid dysphagia before eating?**					
Yes, always	301 (92.9)	10 (66.7)	54 (98.2)	365 (92.6)	
Often	21 (6.5)	3 (20.0)	1 (1.8)	25 (6.4)	
Rarely	1 (0.3)	2 (13.3)	0 (0.0)	3 (0.8)	<0.001
Never	1 (0.3)	0 (0.0)	0 (0.0)	1 (0.2)	
**Do you provide straws or syringes for the administration of liquid topatients with liquid dysphagia?**					
Yes	190 (58.6)	9 (60.0)	42 (76.4)	244 (61.2)	0.044
Not	134 (41.4)	6 (40.0)	13 (23.6)	153 (38.8)	
**Do you inform family members about preventive and bronchoaspiration measures?**					
Yes, always	268 (82.7)	11 (73.3)	44 (80.0)	323 (81.9)	
Often	42 (13.0)	4 (26.7)	4 (7.3)	50 (12.7)	0.093
Rarely	10 (3.1)	0 (0.0)	5 (9.1)	15 (3.8)	
Never	4 (1.2)	0 (0.0)	2 (3.6)	6 (1.5)	
**Have you witnessed a bronchoaspiration?**					
Yes	245 (75.6)	12 (80.0)	44 (80.0)	301 (76.4)	0.736
Not	79 (24.4)	3 (20.0)	11 (20.0)	93 (23.6)	
**What was the most serious result after witnessing bronchoaspiration?**					
No incidents	53 (16.4)	1 (6.7)	19 (34.5)	73 (18.5)	
Pneumonia	153 (47.2)	6 (40.0)	16 (29.1)	175 (44.4)	
Death	42 (13.0)	5 (33.3)	10 (18.2)	57 (14.5)	0.005
I have not witnessed any aspiration	76 (23.5)	3 (20.0)	10 (18.2)	89 (22.6)	

ANCT: Auxiliary Nursing Care Technician, EAT-10: Eating Assessment Tool, MECV-V: Clinical Examination of Volume Viscosity.

## Data Availability

The data are collected in a database prepared by the research team.

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
