# Peer review of "Knowledge and Practice of Health Professionals in the Management of Dysphagia"

_ijerph, 2021, doi:10.3390/ijerph18042139_

Round 1

Reviewer 1 Report

Data from respondents who could not define dysphagia should be excluded. 

Terminology should be more clear and consistent. Ex. screening versus evaluation?  "Dysphagia approach"? vs evaluation and treatment of patients with dysphagia (what approach?)

It should be clearly stated that silent aspiration occurs when there is an absence of a protective response. It isn't just that the person doesn't make a sound, cough, etc. they do not protect themselves.

Recommendation of viscosity is made on more than just severity of dysphagia. Some patients can use strategies, postures, maneuvers to manage a given viscosity. The patient may also be on a free water protocol, etc. 

It should be cleary stated that an instrumental exam is required to objectively diagnose silent aspiration. 

It should not be inferred that a training program will make an RN, MD, or tech capable of performing anything beyond a screening. That said, even trusting an MD or an RN to perform a screening with patients with diagnoses associated with a high percentage of silent aspiration is highly questionable. Under no circumstance should a tech be permitted to perform a screening even on patients without such diagnoses.

A much clearer description of what can result in aspiration is needed.

Suggesting that more training is fine but it should be cleary stated that this training will allow RNs, MDs, and techs to provide daily care of the patient with dysphagia and that they will know when to consult speech pathology and/or when to alert speech pathology and the md team that a patient is exhibiting signs and symptoms during intake of their currently prescribed diet.

Author Response

Dear reviewer,

Firstly, we appreciate the time dedicated to our manuscript, as well as the comments that you ask for that help us to know doubts that a future reader may have, if the manuscript gets published.

Secondly, we answer to the questions that you have made, with aim of resolving doubts raised by our manuscript.

Data from respondents who could not define dysphagia should be excluded. 

Respondents who did not define dysphagia were not excluded as we wanted to value not only the lack of knowledge, but also professional interventions with risk patients. Sometimes, many professionals treat the patients with a diagnosis and cannot define it, however they know how to perform according to the guides.

Terminology should be more clear and consistent. Ex. screening versus evaluation?  "Dysphagia approach"? vs evaluation and treatment of patients with dysphagia (what approach?)

We mean using the term ‘’Dysphagia approach’’ to the holistic management of the patient with dysphagia, which includes assessment and treatment, as well as other interventions. It is consider as a wider term for the management of these patients.

It should be clearly stated that silent aspiration occurs when there is an absence of a protective response. It isn't just that the person doesn't make a sound, cough, etc. they do not protect themselves.

Must be clearly indicated that silent aspiration is produced when there is not a protected response. It is not only that the person does not make any sound, cough, etc., they are not protected.

Recommendation of viscosity is made on more than just severity of dysphagia. Some patients can use strategies, postures, maneuvers to manage a given viscosity. The patient may also be on a free water protocol, etc. 

As you comment, patients with dysphagia can use compensation strategies, postures and maneuvers in order to improve swallowing. In the text, we talk about the importance of diagnosing using adequate instruments and techniques, V-VST in this particular case, to diagnose and prescribe a viscosity and volumen adapted to the current patient’s situations. We understand that the prescribed viscosity and compensation strategies are complementaries and that is the reason why it is questioned if the patient has been accommodated… We have not asked about techniques and maneuvers of compensation as these questions are very specific and firstly we would like to know whether patients were changed to a posture that would make swallowing easier (supine position with an angle below 45º).

It should be cleary stated that an instrumental exam is required to objectively diagnose silent aspiration. 

We have added to the text new information about the need of performing instrumental tests, mainly swallowing videofluiroscopy, in order to diagnose a silent aspiration.

It should not be inferred that a training program will make an RN, MD, or tech capable of performing anything beyond a screening. That said, even trusting an MD or an RN to perform a screening with patients with diagnoses associated with a high percentage of silent aspiration is highly questionable. Under no circumstance should a tech be permitted to perform a screening even on patients without such diagnoses.

We agree with your comment, and our objective was not to infer that dysphagia training make diferent professionals competent to perform its screening or diagnosis. What we try is to know limitations that some professionals may have in the management of these patients, as we can see situations, that even when dysphagia has been diagnosed, the texture is not adapted or the patient has not been accommodated in a proper posture. The role of the technician is important when feeding a dependent patient, and we wanted to know different points that a training programme must include.

Dysphagia screening or diagnosis using clinical tests, prescription and diet adaptation, compensatory measures… must be performed by a qualified professional and highly trained in the field of dysphagia.

In our case, the screening and diagnosis are carried out by a multidisplinary team that forms the dysphagia unit. The same team is the one commissioned to train different professionals for an optimal management of these kind of patients.

A much clearer description of what can result in aspiration is needed.

It has been added to the text new information about aspiration consequences.

Suggesting that more training is fine but it should be cleary stated that this training will allow RNs, MDs, and techs to provide daily care of the patient with dysphagia and that they will know when to consult speech pathology and/or when to alert speech pathology and the md team that a patient is exhibiting signs and symptoms during intake of their currently prescribed diet.

We have added this clarification in conclusions.

Once again, we appreciate the time and attention dedicated to our manuscript. We really hope we have reached your expectations, with the modifications made and that the explanations to those that we have not modified be considered as appropiate.

Kind regards.

Reviewer 2 Report

I believe that the manuscript treats a topic of interest and relevance for the readers of the journal and I have found interesting the results of the manuscript.

Author Response

Dear reviewer,

We appreciate the time dedicated to our manuscript.

Thank you very much for your feedback. It helps us to keep progressing in this field.

Kind regards.

Reviewer 3 Report

Specific comments:

  1. Abstract: At the beginning of the abstract, the authors should clearly state that the manuscript is focused on liquid dysphagia.
  2. Introduction (line 44): The authors should point out that they focus on the liquid dysphagia.
  3. The structure of the Introduction section could be improved, e.g. malnutrition is mentioned in line 54 and again in line 63; some specific values (e.g. volumes in line 86) could be omitted, etc.
  4. Methods: The authors should mention whether the data in paragraph 2 are for Spain, and the study was conducted in the same country (lines 125-128).
  5. The authors should explain the X2 test (is it a general statistic method or is it a name used in the R-commander program only?). Add the producer of the program (line 135).
  6. Results: In the text, use only one decimal place in percentage values (paragraphs 1-3 in the Results).
  7. Check the use of p values in the text: e.g. Is the p value (line 143) necessary for the manuscript?
  8. The whole text should be carefully checked: e.g. remove the dot in Table 2 (question 3, answer 4); check the spelling: questionnaire in the head of Table 2; tolos (line 29); Will (line 37), area (line 96); sherbets (line 203) vs sorbets (line 264), etc.
  9. The English language in the whole text should be improved: e.g. the first word in Abstract is a verb; to inference (line 297), etc. The language is sometimes simple, the same phrases are repeated several times.

Author Response

Dear reviewer,

Firstly, we appreciate the time dedicated to our manuscript, as well as the comments that you ask for that help us to know doubts that a future reader may have, if the manuscript gets published.

Secondly, we answer to the questions that you have made, with aim of resolving doubts raised by our manuscript.

Comment reviewer

Abstract: At the beginning of the abstract, the authors should clearly state that the manuscript is focused on liquid dysphagia.

Response.

We have modified the beginning of the abstract and we have added liquid dysphagia.

Comment reviewer

Introduction (line 44): The authors should point out that they focus on the liquid dysphagia.

Response.

In line 44, we do not focus on liquid dysphagia because it is an introduction of the dysphagia, so we remark that there are liquid and solid dysphagia. In the next paragraph is where we specify that we talk about liquid dysphagia.

Comment reviewer

The structure of the Introduction section could be improved, e.g. malnutrition is mentioned in line 54 and again in line 63; some specific values (e.g. volumes in line 86) could be omitted, etc.

Response.

Line 63 has been added to the previous paragraph, as once dysphagia consequences are generally commented in line 54, we specify a bit more. Regarding malnutrition and dehydration, we mention that fasting is used to avoid bronchoaspiration.

Comment reviewer

Methods: The authors should mention whether the data in paragraph 2 are for Spain, and the study was conducted in the same country (lines 125-128).

Response.

‘’In Spain’’ has been added to specify that data and the study was made in Spain.

Comment reviewer

The authors should explain the X2 test (is it a general statistic method or is it a name used in the R-commander program only?). Add the producer of the program (line 135).

Response.

‘’X2 test’’ is the Chi square test method. It has been modified in the text.

Comment reviewer

Results: In the text, use only one decimal place in percentage values (paragraphs 1-3 in the Results).

Response.

It has been modified paragraph 1-3 and the second decimal has been deleted.

Comment reviewer

Check the use of p values in the text: e.g. Is the p value (line 143) necessary for the manuscript?

Response.

We think that it is important to point p value as the reader can read if there was or not statistical significance and confidence level.

Comment reviewer

The whole text should be carefully checked: e.g. remove the dot in Table 2 (question 3, answer 4); check the spelling: questionnaire in the head of Table 2; tolos (line 29); Will (line 37), area (line 96); sherbets (line 203) vs sorbets (line 264), etc.

The English language in the whole text should be improved: e.g. the first word in Abstract is a verb; to inference (line 297), etc. The language is sometimes simple, the same phrases are repeated several times.

Response.

The text has been modified and the English language reviewed with professional translator support.

Once again, we appreciate the time and attention dedicated to our manuscript. We really hope we have reached your expectations, with the modifications made and that the explanations to those that we have not modified be considered as appropiate.

Kind regards.

Round 2

Reviewer 1 Report

The revisions made the purpose of the study more clear. 

Reviewer 3 Report

There are small errors, which should be corrected during the proof reading.

Decimal points (instead of commas) should be used: lines 17 and 148.

In Table 1, change n (%) to (n, %).